# Meta-analysis of the effectiveness and safety of robotic-assisted versus laparoscopic transabdominal preperitoneal repair for inguinal hernia

**Xi Li, Yue-Juan Li, Hui Dong, Deng-Chao Wang, Jian Wei** [ORCID] *

Department of General Surgery, Zigong Fourth People's Hospital, Zigong, Sichuan, China

* jianweipuwaike@163.com

## Abstract

### Background

Inguinal hernia is a common global disease. This study aims to investigate the effectiveness and safety of robot-assisted transabdominal preperitoneal repair (RTAPP) and laparoscopic transabdominal preperitoneal repair (LTAPP) for inguinal hernia.

### Methods

We conducted a thorough search in Cochrane Library, Embase, and PubMed for relevant clinical studies. After applying inclusion and exclusion criteria, the quality of selected studies was assessed using the Jadad scale for randomized controlled studies and the Newcastle-Ottawa scale for observational studies. Meta-analysis was performed using RevMan 5.3 software.

### Results

A total of ten studies were included, comprising two randomized controlled studies and eight non-randomized controlled studies. Meta-analysis results revealed no statistically significant differences between the RTAPP group and the LTAPP group regarding hospital stay [MD = 0.21 days, 95% CI (-0.09, 0.51), P = 0.17], incidence of seroma [OR = 0.85, 95% CI(0.45, 1.59), P = 0.61], overall complication rate [OR = 1.22, 95% CI(0.68, 2.18), P = 0.51], readmission rate [OR = 1.31, 95% CI(0.23, 7.47), P = 0.76], and recurrence rate [OR = 0.82, 95% CI(0.22, 3.07), P = 0.77]. However, the RTAPP group had longer operation time compared to the LTAPP group [MD = 14.02 minutes, 95% CI (6.65, 21.39), P = 0.0002], and the cost of the RTAPP procedure was higher than that of the LTAPP procedure [MD = $4.17 thousand, 95% CI (2.59, 5.76), P<0.00001].

### Conclusion

RTAPP for inguinal hernia is a safe and feasible approach, however, it is associated with increased operation time and treatment costs.

**Data Availability Statement:** All relevant data are within the manuscript and its Supporting information files.

**Funding:** The author(s) received no specific funding for this work.

**Competing interests:** The authors have declared that no competing interests exist.

## Introduction

Inguinal hernia is a common global disease, and with the aging population, its incidence continues to rise [1]. The most frequently employed techniques for laparoscopic inguinal hernia surgery are transabdominal preperitoneal repair (TAPP) and totally extraperitoneal repair (TEP) [2]. The emergence of robotic surgical systems has led to significant advancements and groundbreaking changes in various medical disciplines, including general surgery, hepatobiliary surgery, urology, obstetrics and gynecology, cardiovascular surgery, thoracic surgery, and pediatric surgery. These systems have played a pivotal role in promoting the development of minimally invasive surgery and have ushered in a revolutionary era [3, 4]. The cutting-edge da Vinci robotic surgical system has gradually gained traction in some countries and regions and is redefining minimally invasive surgery as the third generation of surgical technology, following open surgery and laparoscopy [5, 6]. With its unique features, including 3D imaging, a mechanical arm with 7 degrees of freedom, tremor filtration, and more ergonomic design, da Vinci surgery has elevated surgical precision to unprecedented heights and ushered minimally invasive surgery into a new era [7, 8]. Currently, it has been successfully applied in abdominal wall incisional hernia repair, paraesophageal hernia repair, and inguinal hernia repair, demonstrating its technical advantages [9–11]. While several randomized controlled studies and non-randomized controlled studies have compared RTAPP with LTAPP in the treatment of inguinal hernia [12–21], there is limited data from single-center studies, and clinical reports yield inconsistent results. By specifically focusing on TAPP procedures and including the latest studies from 2023, this meta-analysis aims to provide a comprehensive evaluation of the effectiveness and safety of RTAPP in inguinal hernia treatment. The inclusion of recent studies not only enhances the timeliness of our findings but also contributes novel insights into the evolving landscape of robotic-assisted minimally invasive surgery for inguinal hernia.

## Methods

Our systematic review adheres to the Preferred Reporting Items for Systematic Reviews and Meta-Analyses for Protocols guidelines [22], and it is registered under the number INPLASY202390048.

### Inclusion and exclusion criteria

**Inclusion criteria.** (1) Study Subjects: Individuals diagnosed with inguinal hernia through preoperative physical examination, aged over 18 years, of any gender. (2) Intervention: Either robotic-assisted transabdominal preperitoneal tension-free repair or laparoscopic transabdominal preperitoneal tension-free repair, with no restriction on the type of mesh used during surgery. (3) Study Type: Randomized controlled trials, non-randomized controlled studies (retrospective or case-control studies), limited to publications in English. (4) Outcome Measures: Operation time, hospital stay, cost, incidence of seroma, overall complication rate, readmission rate, recurrence rate.

**Exclusion criteria.** (1) Non-comparative studies. (2) Case reports, abstracts, conference reports, reviews. (3) Studies where surgical procedures did not involve either robotic-assisted transabdominal preperitoneal tension-free repair or laparoscopic transabdominal preperitoneal tension-free repair. (4) Studies from which outcome measures could not be extracted. (5) Literature inaccessible in full text.

### Retrieval strategy

A comprehensive computer-based retrieval was conducted on The Cochrane Library, Embase database, and PubMed database. The retrieval period for all databases extended from their inception to April 7, 2023. The search terms used in the databases were: inguinal hernia, groin hernia, hernioplasty, transabdominal preperitoneal (TAPP), robot, robotic. Additionally, in order to obtain further study on this topic, references from the included literature were also reviewed to determine if they met the inclusion criteria.

### Literature screening and data extraction

Literature retrieval and data extraction were carried out independently by two authors. Any disagreements were resolved through discussion or by seeking assistance from a third researcher. The extracted data included: (1) General information: first author, publication year, country, sample size, gender, age, patch type, follow-up duration; (2) Outcome measures: operation time, hospital stay, cost, incidence of seroma, overall complication rate, readmission rate, recurrence rate.

### Quality assessment

Two authors independently conducted quality assessments of the included studies, with cross-verification. In cases of disagreement during the evaluation process, disagreements were resolved through discussion or adjudicated by a third author. The quality of included randomized controlled studies was assessed using the modified Jadad scale, which includes four criteria: (1) random sequence generation, (2) allocation concealment, (3) blinding, and (4) withdraws and dropouts. The total score is 7 points, with scores of ≤3 considered low-quality literature and scores of 4–7 considered high-quality literature [23]. Non-randomized controlled studies were assessed for quality using the NOS (Newcastle-Ottawa Scale), with scoring criteria including (1) selection, (2) comparability, and (3) exposure. Scores of 7–9 indicate high-quality studies, scores of 4–6 indicate medium-quality studies, and scores of 1–3 indicate low-quality studies [24].

### Statistical analysis

Data from the included literature were combined and analyzed using RevMan 5.3 software. For continuous variables and binary variables in the studies, mean differences (MD) and odds ratios (OR) were calculated as effect measures along with their corresponding 95% confidence intervals (CI). Heterogeneity among the included studies was assessed using the chi-squared ($\chi^2$) test, and the magnitude of heterogeneity was quantified using the $I^2$ statistic. If there was no significant heterogeneity among the studies ($I^2 \leq 50\%$, $P \geq 0.10$), a fixed-effects model was used for analysis. If heterogeneity was present ($I^2 > 50\%$, $P < 0.10$), a random-effects model was employed for analysis. For indicators with more than 10 included studies, the potential publication bias was assessed using a funnel plot of the main results. If the plot showed good symmetry, it indicated no significant publication bias [25]. The significance level was set at $\alpha = 0.05$.

## Results

### Literature search results

Initially, a total of 1,033 articles were retrieved from various databases, and an additional 4 articles were identified through manual searches. After reviewing titles and abstracts, 165 duplicate articles were excluded, along with 803 articles unrelated to the research objectives, 36

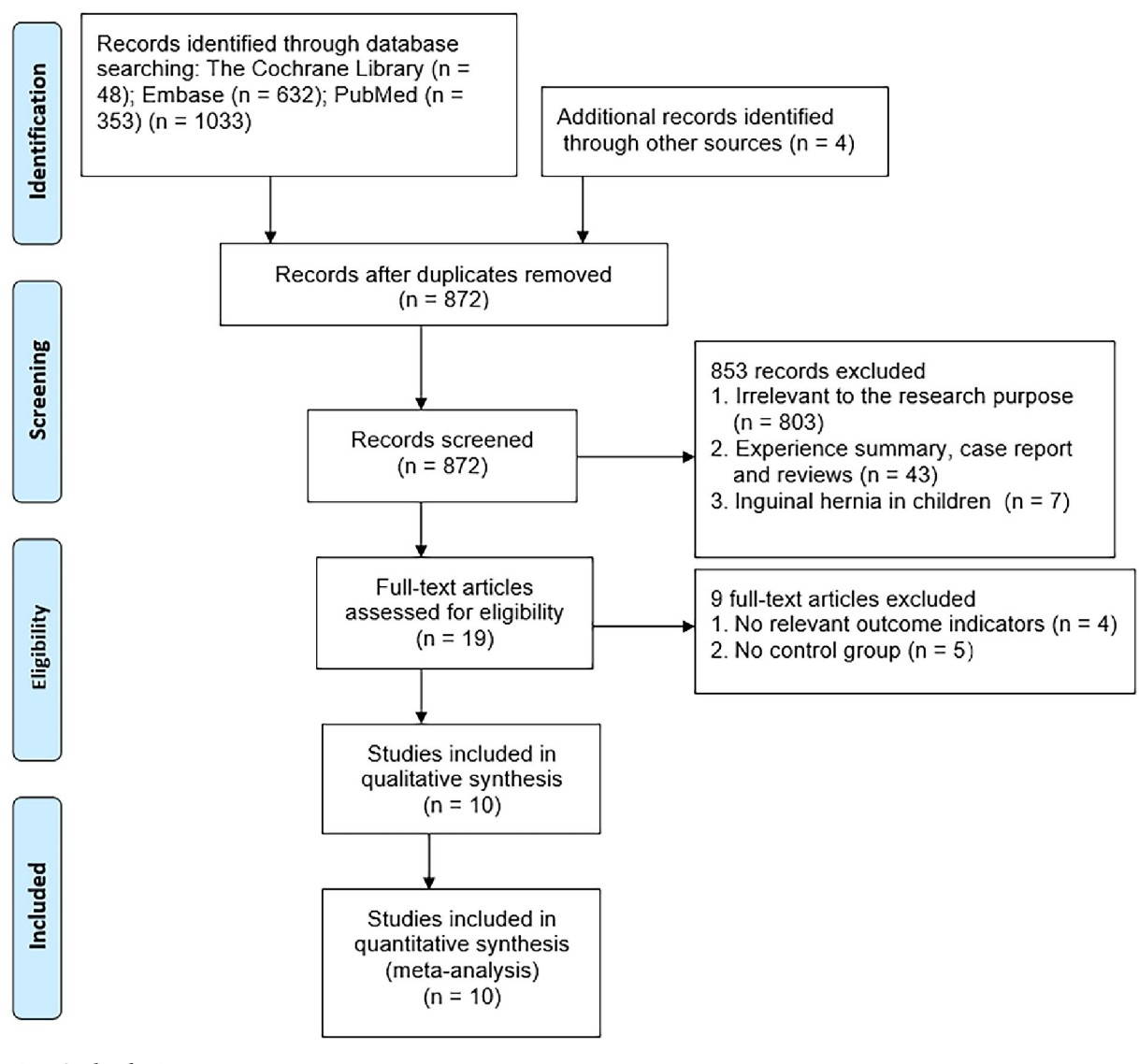

**Fig 1. Study selection.**

articles comprising empirical summaries, reviews, and case reports, and 7 articles pertaining to pediatric inguinal hernias. The remaining 19 articles underwent full-text screening, and 4 articles lacked relevant outcome measures, while 5 articles were excluded due to the absence of control groups. Following the sequential screening process described above, a total of 10 articles [10–29] were ultimately included. A detailed overview of the selection process is shown in Fig 1. The basic characteristics of the included literature are provided in Table 1.

## Results of literature quality assessment

Among the included studies, 2 articles [16, 21] were randomized controlled trials (RCTs) with a score of 4, indicating high quality. Eight articles [12–15, 17–20] were non-randomized controlled studies, and all of them received high-quality ratings based on the Newcastle-Ottawa

**Table 1. Characteristics of the studies included in this meta-analysis.**

| Study and publication year | Country | Group | Sample size (M/F) | Age (years) | Type of mesh | Follow-up time | Score (Jadad/ NOS) | Outcome indicators |
|---|---|---|---|---|---|---|---|---|
| Ayuso 2023 [12] | USA | RTAPP | 141 (Na/Na) | 58.6 ± 13.8 | Midweight polypropylene mesh (3D Max) | 13.0 ±13.3 months | 7 | ①②③⑥⑦ |
| | | LTAPP | 141 (Na/Na) | 54.4 ± 15.5 | Midweight polypropylene mesh (3D Max) | 13.0 ± 13.3 months | | |
| Choi 2023 [13] | Korea | RTAPP | 50 (50/0) | 54.4 ± 14.0 | 15.7×10.3 cm large size mesh (3DMax Light Mesh) | 30 days | 7 | ①②③④⑥⑦ |
| | | LTAPP | 50 (49/1) | 64.4 ± 14.8 | 15.7×10.3 cm large size mesh (3DMax Light Mesh) | 30 days | | |
| Gerdes 2022 [14] | Switzerland | RTAPP | 29 (27/2) | 62 (36–81) | BARD 3D Lightmesh 10×15 cm | 1 year | 8 | ⑤⑥ |
| | | LTAPP | 29 (24/5) | 53 (21–82) | BARD 3D Lightmesh 10×15 cm | 1 year | | |
| Hsu 2022 [15] | USA | RTAPP | 207 (178/29) | 52.0 (38.0–62.0) | Na | 4 weeks | 7 | ① |
| | | LTAPP | 212 (Na/Na) | 57.0 (45.0–67.0) | Na | 4 weeks | | |
| Miller 2023 [16] | USA | RTAPP | 48 (Na/Na) | Na | Polypropylene mesh at least 10 cm×15 cm | 2 years | 4 | |
| | | LTAPP | 54 (Na/Na) | Na | Polypropylene mesh at least 10 cm×15 cm | 2 years | | |
| Muysoms 2018 [17] | Belgium | RTAPP | 49 (48/1) | Na | 12 × 16 cm (Progrip Laparoscopic Self-Fixating Mesh) | 4 weeks | 8 | ④⑤ |
| | | LTAPP | 63 (61/2) | Na | 12 × 16 cm (Progrip Laparoscopic Self-Fixating Mesh) | 4 weeks | | |
| Muysoms 2021 [18] | Belgium | RTAPP | 404 (377/27) | 60.0 ± 61.7 | Self-gripping monofilament polyester mesh | 4 weeks | 7 | ⑤⑥ |
| | | LTAPP | 272 (237/35) | 60.3 ± 62.0 | Self-gripping monofilament polyester mesh | 4 weeks | | |
| Okamoto 2023 [19] | Japan | RTAPP | 80 (76/4) | 70 (61–75) | Self-gripping mesh sized 15×10 cm | Na | 7 | ①④⑦ |
| | | LTAPP | 80 (75/5) | 71 (62.5–76) | Self-gripping mesh sized 15×10 cm | Na | | |
| Peltrini 2023 [20] | Italy | RTAPP | 40 (35/5) | 56 ± 12 | Ultrapro 24, Progrip 10, Polipropilene 6 | 35 ± 8 months | 7 | ①②④⑤⑥⑦ |
| | | LTAPP | 80 (71/9) | 56 ± 14 | Ultrapro 38, Progrip 4, Parietex 36, Polipropilene 2 | 52 ± 14 months | | |
| Prabhu 2020 [21] | USA | RTAPP | 48 (Na/Na) | 56.1 ± 14.1 | Polypropylene mesh at least 10 cm×15 cm | 30 days | 4 | ①③⑥ |
| | | LTAPP | 54 (Na/Na) | 57.2 ± 13.3 | Polypropylene mesh at least 10 cm×15 cm | 30 days | | |

F, female; LTAPP, laparoscopic transabdominal preperitoneal; M, male; Na, Not available; NOS, Newcastle-Ottawa scale; RTAPP, robotic-assisted transabdominal preperitoneal.

① operation time; ② hospital stay; ③ cost; ④ incidence of seroma; ⑤ overall complication rate; ⑥ readmission rate; ⑦ recurrence rate.

Scale (NOS). Among these, 2 articles [14, 17] scored 8 points, while the remaining 6 articles scored 7 points [12, 13, 15, 18–20]. The detailed scoring results are presented in Table 1.

## Meta-analysis results

**Operation time.** Six studies [12, 13, 15, 19–21] reported operation time. There was significant statistical heterogeneity among the studies (P < 0.00001, I2 = 98%). A random-effects model was used to combine the effect sizes for the meta-analysis, which indicated that the RTAPP group had a longer operation time compared to the LTAPP group [MD = 14.02

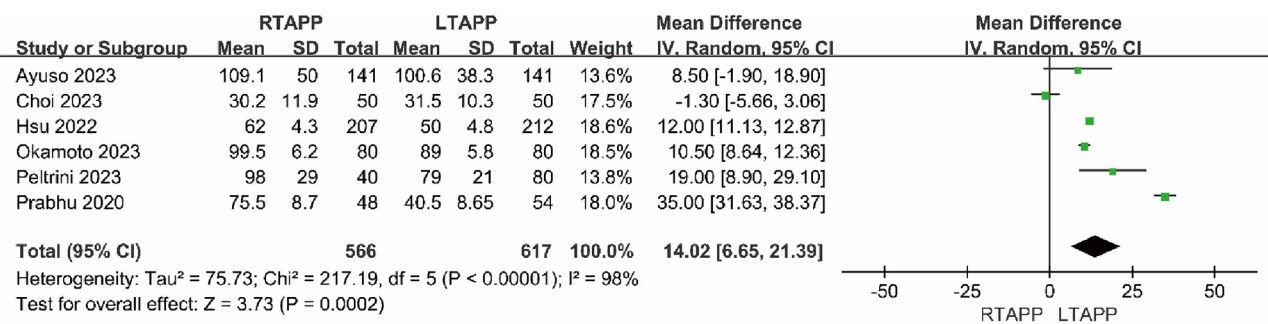

**Fig 2. Forest plot comparing the operation time between the two groups.**

minutes, 95% CI (6.65, 21.39), P = 0.0002], and this difference was statistically significant. Sensitivity analysis, conducted by sequentially excluding individual included studies, showed that the direction of the combined effect value remained unchanged after each exclusion, suggesting the overall stability of the study results, as shown in Fig 2.

**Hospital stay.** Three studies [12, 13, 20] reported hospital stay. There was statistical heterogeneity among the studies (P = 0.05, I2 = 67%). A random-effects model was used to combine the effect sizes for the meta-analysis, which indicated that there was no statistically significant difference in hospital stay between the RTAPP group and the LTAPP group [MD = 0.21 days, 95% CI (-0.09, 0.51), P = 0.17]. Sensitivity analysis, performed by sequentially excluding individual included studies, demonstrated that the direction of the combined effect value remained unchanged after each exclusion, suggesting the overall stability of the study results, as shown in Fig 3.

**Cost.** Three studies [12, 13, 21] reported on cost. There was significant statistical heterogeneity among the studies (P < 0.00001, I2 = 100%). A random-effects model was used to combine the effect sizes for the meta-analysis, which indicated that the cost in the RTAPP group was higher than that in the LTAPP group [MD = $4.17 thousand, 95% CI (2.59, 5.76), P < 0.00001], and this difference was statistically significant. Sensitivity analysis was conducted, and when the study by Choi 2023 [13] or Prabhu 2020 [21] was excluded, the results showed no statistically significant difference in cost between the two groups. Therefore, it suggests that the stability of the results is relatively low, and it is recommended that future researchers conduct more studies on this aspect, as shown in Fig 4.

**Incidence of seroma.** Four studies [13, 17, 19, 20] reported the incidence of seroma. The incidence of seroma in the RTAPP group was 19/219 (8.7%), while in the LTAPP group, it was 26/273 (9.5%). There was no statistical heterogeneity among the studies (P = 0.38, I2 = 2%). A

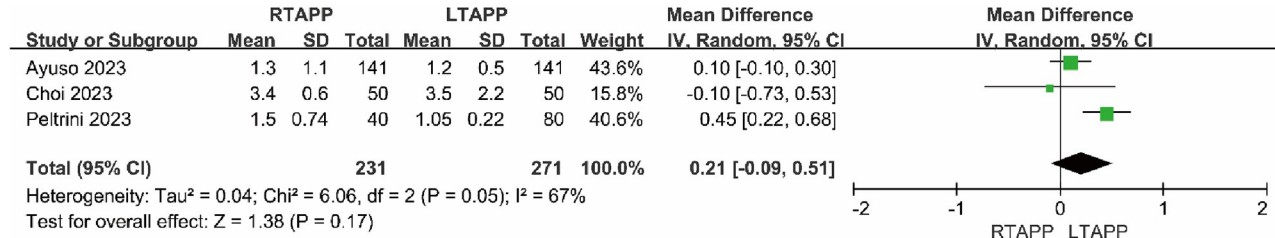

**Fig 3. Forest plot comparing the hospital stay between the two groups.**

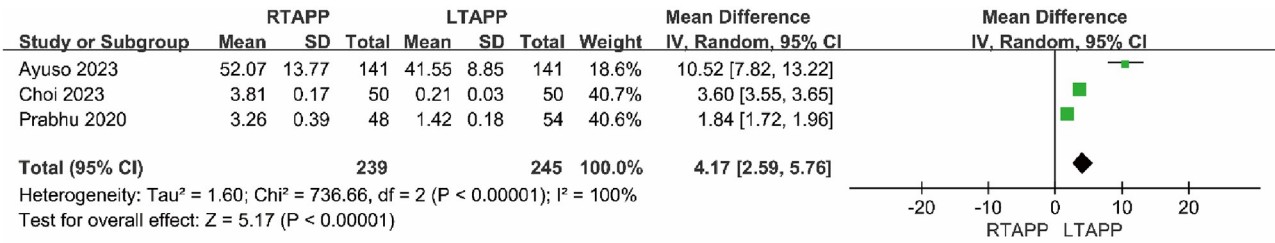

**Fig 4. Forest plot comparing the cost between the two groups.**

fixed-effects model was used to combine the effect sizes for the meta-analysis, which indicated that there was no statistically significant difference in the incidence of seroma between the RTAPP group and the LTAPP group [OR = 0.85, 95% CI (0.45, 1.59), P = 0.61], as shown in Fig 5.

**Overall complication rate.** Four studies [14, 17, 18, 20] reported the overall complication rate. The overall complication rate in the RTAPP group was 28/522 (5.4%), while in the LTAPP group, it was 23/444 (5.2%). There was no statistical heterogeneity among the studies (P = 0.63, I2 = 2%). A fixed-effects model was used to combine the effect sizes for the meta-analysis, which indicated that there was no statistically significant difference in the overall complication rate between the RTAPP group and the LTAPP group [OR = 1.22, 95% CI (0.68, 2.18), P = 0.51], as shown in Fig 6.

**Readmission rate.** Five studies [12–14, 18, 21] reported the readmission rate. The readmission rate in the RTAPP group was 12/672 (1.8%), while in the LTAPP group, it was 11/546 (2.0%). There was statistical heterogeneity among the studies (P = 0.06, I2 = 64%). A random-effects model was used to combine the effect sizes for the meta-analysis, which indicated that there was no statistically significant difference in the readmission rate between the RTAPP group and the LTAPP group [OR = 1.31, 95% CI (0.23, 7.47), P = 0.76]. Sensitivity analysis, performed by sequentially excluding individual included studies, demonstrated that the direction of the combined effect value remained unchanged after each exclusion, suggesting the overall stability of the study results, as shown in Fig 7.

**Recurrence rate.** Five studies [12, 13, 16, 19, 20] reported the recurrence rate. The recurrence rate in the RTAPP group was 3/359 (0.8%), while in the LTAPP group, it was 4/405 (0.9%). There was no statistical heterogeneity among the studies (P = 0.75, I2 = 0%). A fixed-effects model was used to combine the effect sizes for the meta-analysis, which indicated that

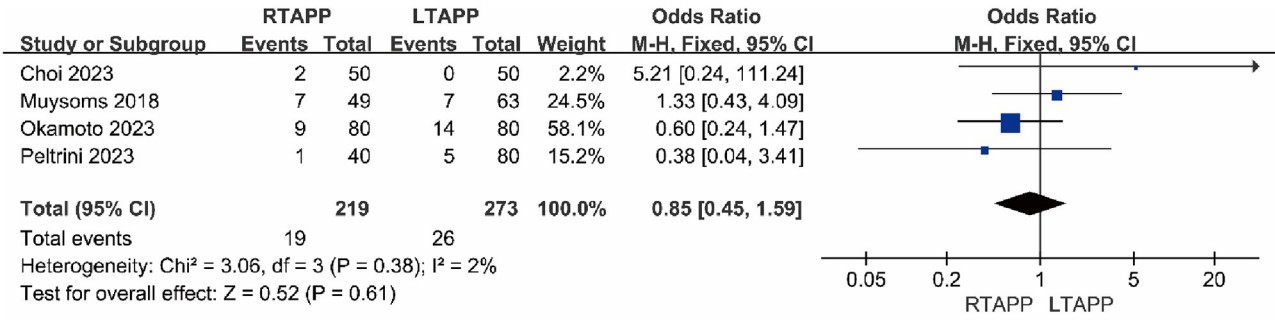

**Fig 5. Forest plot comparing the incidence of seroma between the two groups.**

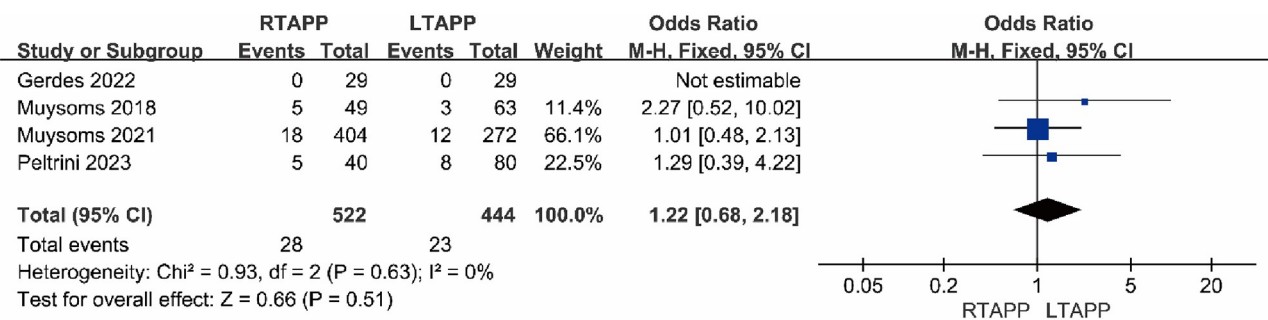

**Fig 6. Forest plot comparing the overall complication rate between the two groups.**

there was no statistically significant difference in the recurrence rate between the RTAPP group and the LTAPP group [OR = 0.82, 95% CI (0.22, 3.07), P = 0.77], as shown in Fig 8.

## Discussion

Inguinal hernia is a common condition in general surgery, with approximately 20 million inguinal hernia repair surgeries performed annually [26]. Traditionally, open and laparoscopic techniques have been the primary methods for inguinal hernia repair [27]. Laparoscopic hernia repair has gained significant attention from surgeons and patient preference due to its advantages, including reduced trauma, lower infection rates, and shorter postoperative recovery times [28]. The introduction of robotic surgical systems has greatly advanced the field of minimally invasive surgery [29]. As robotic technology continues to improve and new surgical instruments are developed, along with surgeons becoming more familiar with the system, its application has gradually expanded to inguinal hernia repair [30]. Currently, research on the comparison between robotic and laparoscopic inguinal hernia repair procedures is limited in sample size, and thus, the feasibility, safety, effectiveness, and cost-effectiveness of robotics in inguinal hernia repair have not been fully determined [12, 31]. Therefore, this study aims to conduct a meta-analysis on the feasibility, safety, effectiveness, and cost-effectiveness of robotic and laparoscopic transabdominal preperitoneal repair in inguinal hernia repair.

This meta-analysis included a total of 10 studies, comprising 2 randomized controlled studies and 8 non-randomized controlled studies. From the findings of this meta-analysis, it is evident that RTAPP has a significantly longer operation time for inguinal hernia compared to

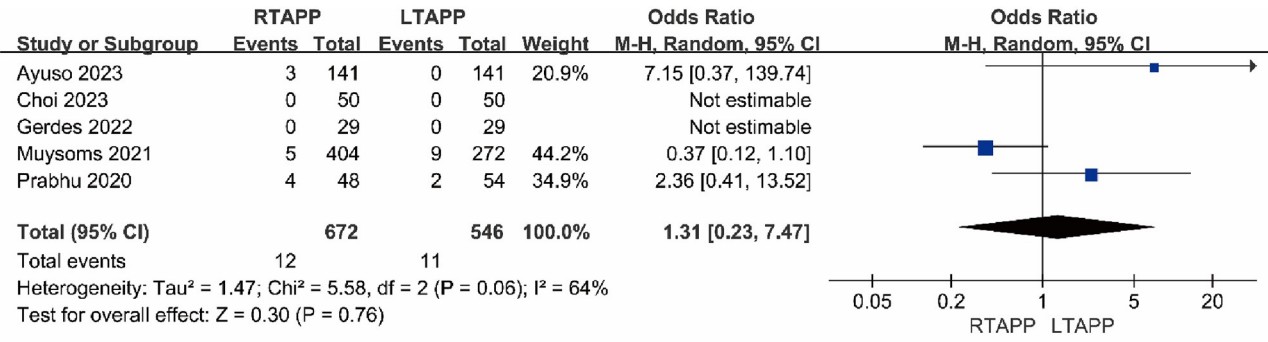

**Fig 7. Forest plot comparing the readmission rate between the two groups.**

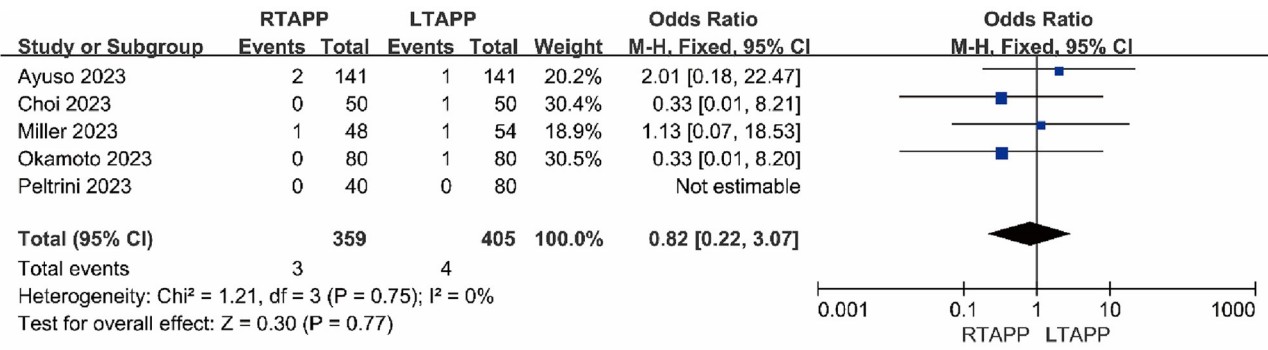

**Fig 8. Forest plot comparing the recurrence rate between the two groups.**

LTAPP. The underlying reasons for this discrepancy may be attributed to two key factors: firstly, the introduction of robotic surgery occurred relatively later, and many of the studies included in this analysis represent early experiences with this technology; secondly, the robotic operating system necessitates a certain amount of preparation time [32]. As surgeons, and even the entire surgical team, become increasingly familiar with the robotic instruments and procedures in the operating room, and as the volume of surgical cases increases, it is possible that the operation time may decrease. In this meta-analysis, three studies reported on cost, with the research conducted in the United States and South Korea. It was found that the cost of robotic surgery was higher than that of laparoscopic surgery, and this difference was statistically significant. While the medical expenses associated with traditional tension-free inguinal hernia repair and laparoscopic hernia repair have gained wide acceptance among patients, the high cost of robotic surgery represents a notable drawback. The primary factors contributing to the cost difference are the expenses associated with medical equipment and the longer operation time [31]. This has, to a certain extent, limited the application of this technology to patients in underdeveloped regions and those without health insurance coverage. However, in the future, as healthcare systems become more comprehensive and operation time decrease, cost may no longer be a significant concern. The anatomical structures in the inguinal region are complex, with numerous blood vessels and nerves [33]. Robotic surgery ensures stable and precise surgical maneuvers [34], allowing for the protection of critical anatomical structures in the inguinal region. The repair of inguinal hernia using RTAPP is deemed safe. This meta-analysis suggests that there is no statistically significant difference between robotic and laparoscopic approaches in terms of overall complication rate, incidence of seroma, and readmission rate. Additionally, it does not lead to an increase in hospital stay, indicating the safety of RTAPP for inguinal hernia. The success of surgery also takes into account the recurrence rate, which is influenced by factors such as the surgeon's experience, mesh size, sufficient overlap of the hernia defect area, surgical site infection, and misdiagnosed hernias [35–37]. In this meta-analysis, the recurrence rate in the RTAPP group was 0.8%, while the LTAPP group had a recurrence rate of 0.9%. There was no statistically significant difference in recurrence rates between the two groups, indicating that RTAPP for inguinal hernia repair does not increase the postoperative recurrence risk and is effective. However, it should be noted that some studies had relatively short follow-up periods, and further research with longer follow-up times is needed to assess recurrence rates more comprehensively in the future.

The strength of evidence in the results of this meta-analysis may be influenced by the following factors: (1) A limited number of included studies with relatively small sample sizes,

further compounded by the scarcity of randomized controlled trials (RCTs) in the available literature. (2) Inclusion of only English-language literature, potentially leading to language bias. (3) There is significant heterogeneity among the included studies in terms of operation time, hospital stay, cost, and readmission rate. This heterogeneity is likely attributed to differences in surgeon expertise, surgical procedures, mesh materials, and fixation methods among the included studies, which inevitably impact the outcomes. (4) Inconsistency in follow-up durations across studies, with short-term follow-up being insufficient to assess and compare hernia recurrence between the two groups. (5) Perceived improved ergonomics and less steep learning curve for robotic inguinal hernia repair may have an impact on the study results and conclusions. (6)Some studies may not have adequately emphasized the importance of randomization, blinding, and allocation concealment in their randomized controlled trials, which, to a certain extent, could affect the strength of evidence in this study.

Comparing RTAPP inguinal hernia repair to LTAPP inguinal hernia repair, it appears to be a safe and viable alternative for the treatment of inguinal hernia, providing a new option. However, it is associated with a longer operation time and higher cost. Due to the limitations of this study, our conclusions still need validation through large-sample, multicenter, rigorously designed, high-quality clinical RCTs.

## Supporting information

**S1 Checklist.**
(DOCX)

## Author Contributions

**Conceptualization:** Xi Li, Hui Dong, Deng-Chao Wang, Jian Wei.

**Data curation:** Xi Li, Yue-Juan Li, Hui Dong.

**Formal analysis:** Xi Li, Yue-Juan Li, Hui Dong.

**Investigation:** Xi Li, Hui Dong.

**Methodology:** Xi Li, Yue-Juan Li, Hui Dong, Deng-Chao Wang, Jian Wei.

**Project administration:** Hui Dong, Jian Wei.

**Resources:** Xi Li, Hui Dong.

**Software:** Xi Li, Yue-Juan Li, Hui Dong, Deng-Chao Wang.

**Supervision:** Hui Dong, Jian Wei.

**Validation:** Xi Li, Yue-Juan Li, Hui Dong, Deng-Chao Wang.

**Visualization:** Xi Li, Yue-Juan Li, Hui Dong, Deng-Chao Wang.

**Writing – original draft:** Xi Li, Hui Dong.

**Writing – review & editing:** Hui Dong, Deng-Chao Wang, Jian Wei.

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
