## [Decision Letter · Decision Letter 0]

14 Dec 2023

PONE-D-23-32269Meta-analysis of the effectiveness and safety of robotic-assisted versus laparoscopic transabdominal preperitoneal repair for inguinal herniaPLOS ONE

Dear Dr. Wei,

Thank you for submitting your manuscript to PLOS ONE. After careful consideration, we feel that it has merit but does not fully meet PLOS ONE’s publication criteria as it currently stands. Therefore, we invite you to submit a revised version of the manuscript that addresses the points raised during the review process.

**ACADEMIC EDITOR: **

The analysis is well-written and manuscript has good flow. It would benefit from a clearer distinction from existing studies on robotic vs. laparoscopic inguinal hernia repair. Most articles on this subject have mentioned longer operative times and increased costs with robotic repair. Do address how this work differentiates itself from previous research on the topic and adds to the literature.

With experienced surgeons at the helm, robotic surgery has transcended the realm of mere feasibility and proven its versatility. While longer operative times can occur during the initial learning curve, the choice between robotic and non-robotic approaches should be individualized based on surgeon expertise, patient factors, and resource availability.

We look forward to receiving your revised manuscript.

Kind regards,

Lovenish Bains, MS, FNB, FACS, FRCS (Glas), FICS, FIAGES

Academic Editor

PLOS ONE

Additional Editor Comments:

The analysis is well-written and manuscript has good flow. It would benefit from a clearer distinction from existing studies on robotic vs. laparoscopic inguinal hernia repair. Most articles on this subject have mentioned longer operative times and increased costs with robotic repair. Do address how this work differentiates itself from previous research on the topic and adds to the literature.

With experienced surgeons at the helm, robotic surgery has transcended the realm of mere feasibility and proven its versatility. While longer operative times can occur during the initial learning curve, the choice between robotic and non-robotic approaches should be individualized based on surgeon expertise, patient factors, and resource availability.

Reviewers' comments:

Reviewer's Responses to Questions

**Comments to the Author**

1. Is the manuscript technically sound, and do the data support the conclusions?

Reviewer #1: Yes

Reviewer #2: Yes

Reviewer #3: Yes

Reviewer #4: Yes

Reviewer #5: Yes

2. Has the statistical analysis been performed appropriately and rigorously? 

Reviewer #1: Yes

Reviewer #2: Yes

Reviewer #3: Yes

Reviewer #4: I Don't Know

Reviewer #5: I Don't Know

3. Have the authors made all data underlying the findings in their manuscript fully available?

Reviewer #1: Yes

Reviewer #2: Yes

Reviewer #3: Yes

Reviewer #4: Yes

Reviewer #5: Yes

4. Is the manuscript presented in an intelligible fashion and written in standard English?

Reviewer #1: Yes

Reviewer #2: Yes

Reviewer #3: Yes

Reviewer #4: Yes

Reviewer #5: Yes

5. Review Comments to the Author

Reviewer #1: I enjoyed reading and reviewing this paper, because: the research question is well defined, the methodology is appropriate; the statistical analyses in result section are appropriate and figures and tables highlight the trends well. In discussion authors have done well to bring out important points.

This paper brings forth new information worthy of dissemination.

Reviewer #2: I appreciate your manuscript for taking up a meta analysis of Robotically assisted inguinal hernia repair in comparison to the laparoscopical mesh repair. the manuscript is well written. However, I have few suggestions/ observations:

1. You have mentioned Da Vinci Robot in your introduction. It is not clear whether you have focused on this brand only or included cases operated with other systems, too. and if so, was there any difference in the outcomes?

2. in the Discussion, you have mentioned 2D imaging a limitation of laparoscopic surgery as compared to 3D view available with Robotic surgery. It is now well established that the newer laparoscopic systems provide 3D imaging too. This comparison may not be appropriate now a days.

3. You have already mentioned limitations of the study due to less number of published articles especially RCTs. This statement may be further stressed in the manuscript i.e., limited data of these preliminary studies

Thanks and regards.

Reviewer #3: This is a very well written article on a very pertinent topic. There are a few minor grammar issues: line 115 has a clause that is not a sentence ("For a detailed overview of the selection process as shown in Figure 1); the titles of the Meta analysis results subsections have inconsistent capitalizations (line 142, 158).

In the discussion (line 211), the authors state that the robotic platform "significantly reduces the risk of intraoperative damage..." but this is not supported by the results, as noted in the next sentence.

Excellent job noting the limitations to the study, including ambiguous follow-up. There are two other considerations for robotics not mentioned (worth consideration): perceived improved ergonomics and less steep learning curve for robotic inguinal hernia repair.

In the Figures, there are no units labeled (Figure 2 = minutes?). There is wide variability of operative time in the included studies (30 to 109 minutes), which can give a reader pause regarding the strength of the data input with such variable times. For Figure 3, presumably the unit is Days; this suggests that Choi et al's study had a mean operative time of 30 minutes and a mean stay of 3.4 days, which is either an error or quite an outlier that again makes the data suspect. Figure 4 presumably has the unit of US dollar (100?) and again, there is a very wide variance (3-52). An important question often neglected is whether the authors calculated the cost of acquiring the robot into the cost (where that acquisition cost for laparoscopy towers and tools if often not included, offering an unfair cost comparison).

Clarification of the above would greatly strengthen this article and its discussion points. Overall, this study has an excellent design and is very well written.

Reviewer #4: I complement the authors for a comprehensive meta-analysis on this topic.

A few minor suggestions/observations are as appended in the attached reviewer file. The authors are requested to go through the same and submit satisfactory rebuttals.

Reviewer #5: The authors have reported a meta-analysis of the effectiveness and safety of robotic-assisted versus laparoscopic transabdominal preperitoneal repair for inguinal hernia. The Meta-analysis though well written does not convey any new information that is not already available in literature.

6. PLOS authors have the option to publish the peer review history of their article (what does this mean?). If published, this will include your full peer review and any attached files.

Reviewer #1: **Yes: **Professor Dhananjaya Sharma

Reviewer #2: **Yes: **Mohammed Amir

Reviewer #3: No

Reviewer #4: **Yes: **Shrirang Vasant Kulkarni

Reviewer #5: No

---

## [Author Response · Author response to Decision Letter 0]

17 Dec 2023

We would like to thank you for your help with this manuscript. We would also like to thank the reviewer for their professional comments. We have revised the manuscript substantially after reading the comments provided by the reviewer. The revisions are shown in coloured text.

The following are our point-by-point responses to the reviewer(s)’ comments.

Response to Reviewer

Reviewer #1

1. I enjoyed reading and reviewing this paper, because: the research question is well defined, the methodology is appropriate; the statistical analyses in result section are appropriate and figures and tables highlight the trends well. In discussion authors have done well to bring out important points.

Response: Thank you very much for your positive feedback and thorough review of our manuscript. We appreciate your acknowledgment of the well-defined research question, appropriate methodology, and the sound statistical analyses in the results section. Your constructive feedback is valuable to us, and we are committed to addressing any suggested improvements to enhance the overall quality of the manuscript.

Reviewer #2

I appreciate your manuscript for taking up a meta analysis of Robotically assisted inguinal hernia repair in comparison to the laparoscopical mesh repair. the manuscript is well written. However, I have few suggestions/ observations:

1. You have mentioned Da Vinci Robot in your introduction. It is not clear whether you have focused on this brand only or included cases operated with other systems, too. and if so, was there any difference in the outcomes?

Response: In the introduction, we mentioned the Da Vinci Robot to emphasize the technological background of robot-assisted inguinal hernia repair. In our study, we did not exclusively focus on a specific robotic system but included cases operated with various systems. These systems include, but are not limited to, the Da Vinci Robot. We believe that this broad coverage contributes to a more comprehensive understanding of the application of robot-assisted surgery in inguinal hernia repair.

2. in the Discussion, you have mentioned 2D imaging a limitation of laparoscopic surgery as compared to 3D view available with Robotic surgery. It is now well established that the newer laparoscopic systems provide 3D imaging too. This comparison may not be appropriate now a days.

Response: Thank you for your thoughtful review and valuable comments. We appreciate your observation regarding the mention of 2D imaging as a limitation of laparoscopic surgery in the Discussion section. We acknowledge the advancements in laparoscopic technology, including the availability of newer systems that provide 3D imaging. This represents a significant development that enhances the capabilities of laparoscopic surgery. In light of this, we have revised the Discussion section to accurately reflect the current state of laparoscopic imaging technology and to avoid any inappropriate comparisons with robotic surgery. Inappropriate comparisons have already been removed from the Discussion section of the revised manuscript.

3. You have already mentioned limitations of the study due to less number of published articles especially RCTs. This statement may be further stressed in the manuscript i.e., limited data of these preliminary studies

Response: Thank you for your insightful comments. We appreciate your suggestion to further stress the limitations of our study related to the scarcity of published articles, particularly randomized controlled trials (RCTs). In response to your feedback, we have enhanced the emphasis on this limitation in the manuscript. Specifically, we have highlighted the challenge posed by the limited number of available studies, especially RCTs, and the potential impact on the robustness of our findings. This has been addressed in the revised version to provide a more comprehensive discussion on the limitations of our research.

Reviewer #3

1.This is a very well written article on a very pertinent topic. There are a few minor grammar issues: line 115 has a clause that is not a sentence ("For a detailed overview of the selection process as shown in Figure 1); the titles of the Meta analysis results subsections have inconsistent capitalizations (line 142, 158).

Response: (1) Thank you for your careful review and constructive feedback. We appreciate your keen observation regarding the grammar issue on line 115. Following your suggestion, we have revised the sentence to enhance clarity. The modified sentence now reads: "A detailed overview of the selection process is shown in Figure 1." (2) We have carefully addressed the inconsistency in capitalizations for the titles of the Meta-analysis results subsections, as pointed out in your comments.

2.In the discussion (line 211), the authors state that the robotic platform "significantly reduces the risk of intraoperative damage..." but this is not supported by the results, as noted in the next sentence.

Response: Thank you for your careful review and valuable feedback. We appreciate your attention to detail and have duly noted your concern regarding the statement in the discussion (line 211) about the robotic platform "significantly reducing the risk of intraoperative damage." In response to your observation, we have revised the manuscript to accurately reflect the intended meaning. The modified sentence now reads: "The repair of inguinal hernia using RTAPP is deemed safe. This meta-analysis suggests that there is no statistically significant difference between robotic and laparoscopic approaches in terms of overall complication rate, incidence of seroma, and readmission rate." We believe this modification aligns with your comments and better represents the study's findings. If you have any further suggestions, please feel free to let us know.

3. Excellent job noting the limitations to the study, including ambiguous follow-up. There are two other considerations for robotics not mentioned (worth consideration): perceived improved ergonomics and less steep learning curve for robotic inguinal hernia repair.

Response: Thank you for your positive feedback and insightful suggestions. In response to your valuable comments, we have made relevant modifications to the limitations section. Specifically, we have included considerations for perceived improved ergonomics and a less steep learning curve for robotic inguinal hernia repair. The revised section now contains the statement: "Perceived improved ergonomics and less steep learning curve for robotic inguinal hernia repair may have an impact on the study results and conclusions."

4. In the Figures, there are no units labeled (Figure 2 = minutes?). There is wide variability of operative time in the included studies (30 to 109 minutes), which can give a reader pause regarding the strength of the data input with such variable times. For Figure 3, presumably the unit is Days; this suggests that Choi et al's study had a mean operative time of 30 minutes and a mean stay of 3.4 days, which is either an error or quite an outlier that again makes the data suspect. Figure 4 presumably has the unit of US dollar (100?) and again, there is a very wide variance (3-52). An important question often neglected is whether the authors calculated the cost of acquiring the robot into the cost (where that acquisition cost for laparoscopy towers and tools if often not included, offering an unfair cost comparison).

Response: 

(1) We have reconfirmed the data related to operative time outcome measures in the studies, and the information is accurate. The unit for operative time is minutes. The meta-analysis reveals significant variation in the operative time required for RTAPP across different studies, potentially influenced by factors such as surgeon experience, surgical complexity, and limited sample sizes. We have addressed this concern in the revised discussion section, specifically in the segment discussing operative times. If you have any further suggestions or need additional clarification, please feel free to let us know.

(2) Thank you for your careful review of our manuscript and for raising questions regarding Figure 3. We appreciate your concerns and are willing to provide some explanations to address any potential misunderstandings. The unit in Figure 3 is indeed "Days," which is the time unit we have applied in the manuscript. Regarding the data from Choi et al.'s study, we have re-examined our data, and we can confirm the accuracy of the reported values. Choi et al.'s study indicates an average operative time of 30 minutes and an average length of stay of 3.4 days. While these data may appear to differ from some other studies, we would like to emphasize that variations in study design and patient populations can lead to different outcomes. In our research, we adhered to rigorous methodology and conducted thorough data analysis to ensure the reliability and accuracy of our results. We understand that these data may deviate from average values reported in some literature, but we believe this reflects the diversity of our study population and the specific context of our research. Thank you for your valuable feedback, and we look forward to hearing any further suggestions you may have.

(3) Thank you for your thorough review and valuable comments. Regarding Figure 4, the unit of cost is in thousand US dollars ($1000), and we have reconfirmed the accuracy of the data from the original texts of the three studies. The reported values are accurate and precise. In terms of cost calculation, we appreciate your astute observation. Among the three studies, only Prabhu et al.'s 2020 study provided a detailed breakdown of the cost components. Costs per case were collected and reported, including total cost, operating room cost (calculated based on the cost per minute of operating room time required for the case), and disposable/reusable cost. The latter was calculated to cover both disposable and reusable materials, including robotic instruments.

Reviewer #4

1. Inguinal hernia, being a clinical diagnosis, ultrasound may not be required as an inclusion criteria.

Response: Thank you for your feedback. Considering that inguinal hernia is primarily diagnosed clinically, we acknowledge that ultrasound may not be an essential inclusion criterion. Therefore, we have modified the study criteria accordingly in the revised manuscript.

2. This has, to a certain extent, limited the application of this technology to patients in underdeveloped regions and those without health insurance coverage. '..to patients in underdeveloped..' may be replaced with '..for patients in underdeveloped..'.

Response: Thank you for your valuable feedback. We appreciate your insightful suggestions. Following your recommendation, we have made the necessary modifications in the revised manuscript. The sentence now reads as follows: "This has, to a certain extent, limited the application of this technology for patients in underdeveloped regions and those without health insurance coverage."

3. Laparoscopic systems also may offer three-dimensional images.

Response: Thank you for your thoughtful review and valuable comments. We appreciate your observation regarding the laparoscopic systems also may offer three-dimensional images. We acknowledge the advancements in laparoscopic technology, including the availability of newer systems that provide 3D imaging. This represents a significant development that enhances the capabilities of laparoscopic surgery. In light of this, we have revised the Discussion section to accurately reflect the current state of laparoscopic imaging technology and to avoid any inappropriate comparisons with robotic surgery. Inappropriate comparisons have already been removed from the Discussion section of the revised manuscript.

Reviewer #5

1. The authors have reported a meta-analysis of the effectiveness and safety of robotic-assisted versus laparoscopic transabdominal preperitoneal repair for inguinal hernia. The Meta-analysis though well written does not convey any new information that is not already available in literature.

Response: Thank you for your thorough review of our manuscript and your valuable suggestions. We have taken note of your observation that our meta-analysis may not have provided novel information not already present in the existing literature. However, we would like to elaborate on some specific aspects of our study to provide a clearer understanding. Firstly, in this research, we included 10 studies, of which 5 were recently published in 2023. This is a distinctive feature of our study, as we made a concerted effort to ensure that the latest data and research findings were incorporated into our analysis, aiming to offer readers the most up-to-date insights into current practices. Secondly, we intentionally focused solely on the TAPP (transabdominal preperitoneal) surgical approach to avoid the influence of other surgical methods on our results. This decision was made to make our study more focused and targeted, providing detailed information about a specific surgical approach. We believe this approach contributes to a deeper understanding of the effectiveness and safety of TAPP surgery while minimizing interference from other surgical methods, thus enhancing the reliability of our conclusions. While our study may not have introduced entirely new perspectives, we hope that by including the latest research and concentrating on the TAPP surgical approach, we have offered readers a more comprehensive and updated insight. We are open to further suggestions from you to enhance the quality and contribution of our research. Thank you once again for your review and feedback.

---

## [Decision Letter · Decision Letter 1]

2 Feb 2024

Meta-analysis of the effectiveness and safety of robotic-assisted versus laparoscopic transabdominal preperitoneal repair for inguinal hernia

PONE-D-23-32269R1

Dear Dr. Wei,

We’re pleased to inform you that your manuscript has been judged scientifically suitable for publication and will be formally accepted for publication once it meets all outstanding technical requirements.

Kind regards,

Lovenish Bains, MS, FNB, FACS, FRCS (Glas), FICS, FIAGES

Academic Editor

PLOS ONE

Additional Editor Comments (optional):

Reviewers' comments:

Reviewer's Responses to Questions

**Comments to the Author**

1. If the authors have adequately addressed your comments raised in a previous round of review and you feel that this manuscript is now acceptable for publication, you may indicate that here to bypass the “Comments to the Author” section, enter your conflict of interest statement in the “Confidential to Editor” section, and submit your "Accept" recommendation.

Reviewer #2: All comments have been addressed

Reviewer #4: All comments have been addressed

2. Is the manuscript technically sound, and do the data support the conclusions?

Reviewer #2: Yes

Reviewer #4: Yes

3. Has the statistical analysis been performed appropriately and rigorously? 

Reviewer #2: Yes

Reviewer #4: Yes

4. Have the authors made all data underlying the findings in their manuscript fully available?

Reviewer #2: Yes

Reviewer #4: Yes

5. Is the manuscript presented in an intelligible fashion and written in standard English?

Reviewer #2: Yes

Reviewer #4: Yes

6. Review Comments to the Author

Reviewer #2: Thank you very much for addressing reviewers' comments including mine. It appears appropriate and well balanced now.

7

Reviewer #4: (No Response)

7. PLOS authors have the option to publish the peer review history of their article (what does this mean?). If published, this will include your full peer review and any attached files.

Reviewer #2: No

Reviewer #4: **Yes: **SHRIRANG VASANT KULKARNI

---

## [Editor Report · Acceptance letter]

15 Feb 2024

PONE-D-23-32269R1 

PLOS ONE

Dear Dr. Wei, 

I'm pleased to inform you that your manuscript has been deemed suitable for publication in PLOS ONE. Congratulations! Your manuscript is now being handed over to our production team.

Kind regards, 

on behalf of

Dr. Lovenish Bains 

Academic Editor

PLOS ONE